# Rural–Urban Disparities in Treatment and Disease-Specific Survival for Patients with Intrahepatic Cholangiocarcinoma: A Retrospective Cohort Analysis of the 2000 to 2021 SEER Database

**DOI:** 10.3390/medsci13030158

**Published:** 2025-09-01

**Authors:** Odelia H. Moon, Mitchell A. Taylor, Omar Hamadi, Aditya Sharma, Peter Silberstein

**Affiliations:** 1School of Medicine, Creighton University, Omaha, NE 68178, USA; odeliamoon@creighton.edu; 2Department of Internal Medicine, Advocate Illinois Masonic Medical Center, Chicago, IL 60657, USA; mitchelltaylor.research@gmail.com (M.A.T.); omar.hamadi@aah.org (O.H.); 3Department of Hematology and Oncology, Geisel School of Medicine at Dartmouth, Lebanon, NH 03755, USA; adityasharma96@hotmail.com; 4Department of Internal Medicine, Division of Hematology and Oncology, Creighton University Medical Center, Omaha, NE 68124, USA

**Keywords:** intrahepatic cholangiocarcinoma, SEER database, rural disparities, health disparities

## Abstract

**Background:** Intrahepatic cholangiocarcinoma (ICC) is an aggressive malignancy with very poor survival. Prior research suggests rural–urban disparities on a regional scale. We aimed to elucidate these disparities in treatment and disease-specific survival (DSS) for ICC patients on a national scale using the SEER database. **Methods:** The SEER database was queried to identify biopsy-confirmed cases of ICC from 2000 to 2021. Differences in clinicopathologic features and treatment between rural and urban patients were assessed using Chi-square and Fischer’s exact tests. Disease-specific survival was compared using Kaplan–Meier and log-rank tests as well as multivariable Cox regressions. **Results:** A total of 14,940 ICC patients were identified. Rural patients were less likely than urban patients to receive chemotherapy (789 of 1588 [49.7%] vs. 7112 of 13,352 [53.3%], *p* = 0.006) and surgical treatment (305 of 1588 [19.2%] vs. 2922 of 13,352 [21.9%], *p* = 0.013). Rural patients experienced reduced 5- and 10-year DSS rates (7.0% and 4.0%) compared to urban patients (9.0% and 6.0%, *p* < 0.001). In multivariable analysis, rural residence independently demonstrated a 17% increased risk of disease-specific mortality compared to their urban counterparts (aHR 1.17, 95% CI 1.03–1.32). **Conclusions:** This study demonstrates significant rural–urban disparities in ICC treatment and survival throughout the US, independent of other prognostic factors. Further investigation into factors driving these disparities is warranted to improve outcomes for rural ICC patients.

## 1. Introduction

Intrahepatic cholangiocarcinoma (ICC) is the second most common primary hepatic malignancy worldwide, following hepatocellular carcinoma, and is associated with a poor prognosis [1]. By the time symptoms such as clinical jaundice and weight loss manifest, the disease is often at an advanced stage, with fewer than 10% of patients achieving a five-year overall survival (OS) rate [1,2]. Additionally, in the United States (US) alone, age-adjusted incidence rates increased by 143% between 2000 and 2017, with a median survival of just 8 months [3,4]. Surgical resection remains the cornerstone of treatment for long-term survival, with the recent addition of postoperative systemic chemotherapy using capecitabine. However, only 25–30% of patients are eligible for surgery and overall surgical treatment rates remain alarmingly low in both rural and urban populations [2,5,6]. These low treatment rates may, in part, be due to the advanced stage at which most cases are diagnosed and the patient’s medical background but may also reflect broader systemic barriers to healthcare access. 

Over the past several decades, various studies have demonstrated persistent rural–urban disparities across the US, significantly impacting outcomes for various cancers [4,7,8]. However, the available literature on such disparities relating to ICC remains limited. Retrospective cohort studies using data from the Texas and Iowa Cancer Registries found that, while patients residing in low-income areas were less likely to receive treatment and experienced reduced OS, rural residence itself was not associated with differences in time to treatment, OS, and disease-specific survival (DSS) [5,6]. Furthermore, a national database study from 2011 to 2015 reported reduced OS among patients with disadvantaged socioeconomic indicators (i.e., MHI < USD 5300, lack of insurance, and unmarried status) [7]. Given these findings and the limited temporal and geographic scope of existing studies, we aimed to investigate if similar patterns are observed at a national level over an extended period of time.

## 2. Methods

The SEER database was queried utilizing SEER*Stat version 8.4.4 to identify patients diagnosed with biopsy-proven cases of ICC (International Classification of Disease for Oncology 3rd edition histology codes 8160/3; primary site codes C22.0–22.1) from 2000 to 2021. Recorded demographic variables of interest included age at diagnosis, sex, race and ethnicity, annual income, and rural–urban living. Rural and urban residence classification was determined using the SEER Rural–Urban Continuum Codes, initially developed by the United States Department of Agriculture’s (USDA) Economic Research Service. Clinicopathologic variables of interest included disease stage at diagnosis, tumor grade, receipt of surgical intervention, chemotherapy, and radiation therapy. The SEER database employs a historical staging system that classifies malignancies into three categories: localized, regional, and distant. According to SEER, localized disease indicates that the malignancy remains confined to its organ of origin without any evidence of spread. Regional stage includes tumors that have extended beyond the primary organ’s boundaries or have involved regional lymph nodes. Distant stage represents the spread of malignancy to remote areas of the body, whether through direct extension, noncontiguous spread to distant organs or tissues, or dissemination via the lymphatic system to distant lymph nodes. This staging system is applicable to multiple malignancy types and accounts for periods when formal staging criteria may have varied. Patient vital status, disease-specific cause of death, and survival in years were also collected. Statistical analysis was completed using SPSS version 29.0.2 (IBM Corp., Armonk, NY, USA) and included Chi-square and Fisher’s exact tests, Kaplan–Meier and log-rank tests, and multivariable Cox regressions. Statistical significance was considered *p* < 0.05. 

## 3. Results

A total of 14,940 ICC patients were identified (Table 1). The overall age-adjusted incidence was 8.6 per one-million person years (95% CI 8.5–8.8), and there was a significant increase in incidence from 2000 to 2021 (annual percent change 6.6; *p* < 0.001). The majority of patients were aged 60–69 (31.4%), male (50.5%), non-Hispanic White (62.9%), had an annual income of USD 75,000+ (59.2%), and resided in urban communities (89.4%). When comparing rural and urban ICC patients, there was a significant association between rural–urban living and annual income (*p* < 0.001), where a greater number of urban patients had an annual income of USD 75,000+ (65.2%) compared to their rural counterparts (8.9%). Significant associations were also observed between rural–urban residence and both chemotherapy (*p* = 0.006) and surgical intervention (*p* = 0.013); compared to urban patients, rural patients were less likely to receive chemotherapy (49.7% vs. 53.3%) and surgical treatment (19.2% vs. 21.9%). There were no significant associations observed between rural–urban living and disease stage (*p* = 0.282), tumor grade (*p* = 0.088), and receipt of radiation therapy (*p* = 0.134).

For survival, the median follow-up time for the entire cohort was 0.58 years (IQR 0.17–1.50) and the 5- and 10-year DSS rates were 9.0% and 6.0%, respectively. Comparing rural and urban ICC patients, rural patients experienced significantly reduced 5- and 10-year DSS rates (7.0% and 4.0%) compared to their urban counterparts (9.0% and 6.0%) (*p* < 0.001) (Figure 1). Multivariable analysis adjusting for age at diagnosis, sex, race and ethnicity, annual income, disease stage at presentation, and tumor grade revealed that rural patients independently experienced a 17% increased risk of disease-specific mortality (aHR 1.17; 95% CI 1.03–1.32) compared to urban patients (Table 2).

## 4. Discussion

In this study, our findings demonstrate persistent rural–urban disparities in both treatment receipt and DSS, with rural patients less likely to undergo chemotherapy or surgical resection and independently experiencing a higher risk of disease-specific mortality compared to their urban counterparts. This aligns with region-specific studies mentioned prior which found significant associations between decreased treatment utilization and lower median household income and survival outcomes [5,6]. A multivariable analysis of 740 Oregon State Cancer Registry patients with ICC treated at referral centers showed rural patients were less likely to receive curative-intent surgery (aOR 4.51; 95% CI 2.89–7.01) or radiation (aOR 2.12; 95% CI 1.26–3.56) and had inferior OS (aHR 0.79; 95% CI 0.62–0.99) compared to those treated at non-referral centers [9]. Furthermore, greater distance to a referral center was independently associated with non-referral center treatment. Our results, leveraging an independent risk factor after adjusting for prognostic covariates like disease stage, tumor grade, and income level, suggest additional factors specific to rural residency may contribute to the observed disparities.

In addition, the reduced rates of chemotherapy and surgery among rural patients highlight a potential gap in treatment accessibility. These differences may be attributable to systemic barriers including fewer specialized hepatobiliary centers in rural areas, limited access to oncology services, socioeconomic challenges, and transportation difficulties [10]. Suraju et al. found that, though rural residency was not an independent predictor of DSS in their multivariable analysis, they observed that rural residents traveled longer distances (>50 miles) to access definitive care [5]. Rural health systems may also have reduced availability of multidisciplinary care teams (MDTs), which are often essential in managing complex cancers such as ICC [5,6,11]. Though no standardized guideline for MDTs in cholangiocarcinomas exists, the European Network for the Study of Cholangiocarcinoma highlighted the necessity of an MDT coordinator, weekly meetings, and a core group of specialists (i.e., surgeon, radiologist, hepatologist, pathologist, endoscopist, gastroenterologist). Thus, the potential absence or inconsistent implementation of MDTs in rural healthcare systems could significantly hinder optimal treatment planning and contribute to the observed disparities for ICC patients in our study.

Interestingly, there were no significant differences in tumor stage or grade at diagnosis between rural and urban patients, suggesting that diagnostic delay may not be the primary driver of worse outcomes in rural populations. Instead, disparities in treatment delivery and follow-up care likely play a more substantial role. This aligns with the observation in the Oregon study that treatment at referral centers—often less accessible in rural areas—was associated with better outcomes, implying that access to optimal treatment rather than stage at presentation may be a key determinant [9]. Furthermore, although rural patients were more likely to fall into lower income brackets, disparities in DSS persisted even after adjusting for income, highlighting the multifactorial nature of geographic disparities in cancer care.

This study has several limitations inherent to use of a cancer registry database. The retrospective nature of the analysis limits causal inference and unmeasured confounders such as insurance information, performance status, and patient preferences that may influence treatment decisions and outcomes. Additionally, although a large national dataset was utilized, granular data on treatment-specific variables such as provider availability, distance to care, health behavior like smoking, and patient-level barriers to accessing treatment were not available. The reliance of a registry database limited our ability to assess the impact of distance to care and availability of specialized centers, factors that were shown to influence treatment decisions and outcomes in rural ICC patients [10].

Despite these limitations, our findings underscore the urgent need for targeted interventions to reduce disparities in ICC outcomes. These could include expanding access to specialized oncology care in rural regions, strengthening referral networks to high-volume centers, and investing in telehealth infrastructure to support continuity of care [11]. Future studies should aim to explore the underlying mechanisms contributing to rural–urban disparities in greater detail, including qualitative assessments of patient and provider perspectives, and evaluate interventions that promote equitable treatment access for all ICC patients.

## Figures and Tables

**Figure 1 medsci-13-00158-f001:**
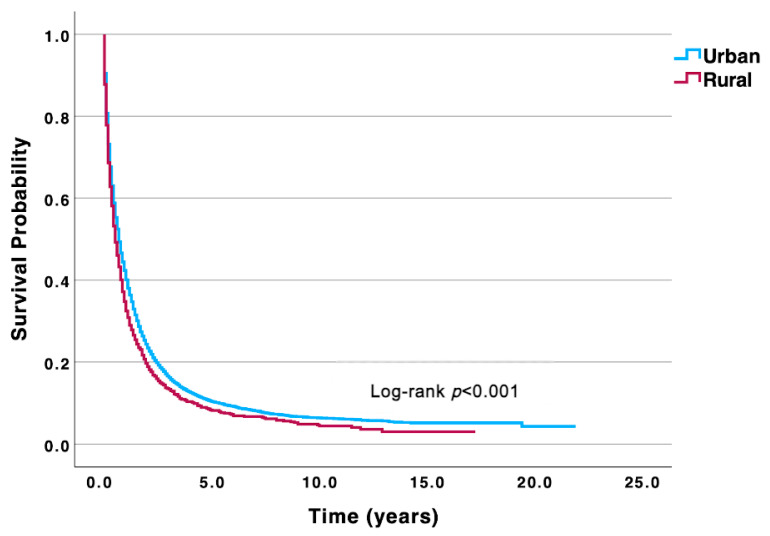
Univariable Kaplan–Meier analysis highlighting significantly reduced DSS in rural ICC patients compared to their urban counterparts. ICC, intrahepatic cholangiocarcinoma; DSS, disease-specific survival.

**Table 1 medsci-13-00158-t001:** Comparison of clinicopathological features between rural and urban ICC patients.

Total n = 14,940	Rural (n = 1588)	Urban (n = 13,352)	*p*-Value
**Age at diagnosis (years)**			0.082 ^⛛^
<40	32 (2.0%)	379 (2.8%)	
40–49	88 (5.5%)	877 (6.6%)	
50–59	285 (17.9%)	2546 (19.1%)	
60–69	524 (33.0%)	4169 (31.2%)	
70–79	458 (28.8%)	3632 (27.2%)	
80+	201 (12.7%)	1749 (13.1%)	
**Sex**			0.176 °
Male	828 (52.1%)	6718 (50.3%)	
Female	760 (47.9%)	6634 (49.7%)	
**Race and ethnicity**			<0.001 ^⛛^
NH White	1341 (84.6%)	8044 (60.4%)	
NH Black	99 (6.2%)	1079 (8.1%)	
NH API	47 (3.0%)	1772 (13.3%)	
NH AIAN	22 (1.4%)	77 (0.6%)	
Hispanic (any race)	77 (4.9%)	2355 (17.7%)	
**Annual income ∞**			**<0.001** °
<USD 74,999 (lower income)	1447 I (91.1%)	4648 (34.8%)	
USD 75,000+ (higher income)	141 (8.9%)	8704 (65.2%)	
**Disease stage at presentation δ**			0.282 ^⛛^
Localized	411 (30.7%)	3332 (28.7%)	
Regional	352 (26.3%)	3149 (27.1%)	
Distant	574 (42.9%)	5146 (44.3%)	
**Tumor grade**			0.088 ^⛛^
Well differentiated (I)	53 (12.0%)	410 (10.0%)	
Moderately differentiated (II)	193 (43.6%)	1878 (45.8%)	
Poorly differentiated (III)	186 (42.0%)	1764 (43.0%)	
Undifferentiated (IV)	11 (2.5%)	51 (1.2%)	
**Surgical intervention ***	301 (19.2%)	2911 (21.9%)	**0.013** °
**Chemotherapy ***	1588 (49.7%)	13352 (53.3%)	**0.006** °
**Radiation therapy ***	242 (15.4%)	1851 (14.0%)	0.134 °

Significant *p*-values (<0.05) are in bold. ^⛛^ Test statistic calculated Chi-square test. ° Test statistic calculated using Fisher’s exact test. AIAN, American Indian/Alaska Native; API, Asian or Pacific Islander; ICC, intrahepatic cholangiocarcinoma; NH, non-Hispanic; SEER, Surveillance Epidemiology and End Results. ***** In comparison to not receiving this treatment. **∞** Annual income data obtained from United States Census Bureau 5-year estimates. **δ** The SEER historical staging system categorizes malignancies into three groups: localized (restricted to the organ of origin), regional (spreading beyond the originating organ or involving nearby lymph nodes), and distant (having spread to a distant part of the body through direct extension, non-contiguous metastasis, or via the lymphatic system to distant lymph nodes). This classification framework is applicable across a broad spectrum of malignancies and accommodates variations in formal staging systems over time.

**Table 2 medsci-13-00158-t002:** Multivariable Cox regression identifying factors associated with disease-specific mortality in ICC patients.

Total n = 14,940	aHR ‡	95% CI	*p*-Value
**Age at diagnosis (years)**			
<40	Reference		
40–49	0.87	0.70–1.09	0.227
50–59	1.05	0.86–1.28	0.656
60–69	1.10	0.91–1.34	0.322
70–79	1.32	1.08–1.60	**0.006**
80+	2.19	1.77–2.72	**<0.001**
**Sex**			
Male	Reference		
Female	0.80	0.75–0.86	**<0.001**
**Race and ethnicity**			
NH White	Reference		
NH Black	1.29	1.13–1.48	**<0.001**
NH API	1.03	0.92–1.15	0.593
NH AIAN	0.99	0.66–1.48	0.952
Hispanic (any race)	1.10	1.00–1.21	0.061
**Annual income**			
<USD 74,999 (lower income)	Reference		
USD 75,000+ (higher income)	0.90	0.84–0.97	**0.006**
**Rural–urban living**			
Urban	Reference		
Rural	1.17	1.03–1.32	**0.014**
**Disease stage at presentation**			
Localized	Reference		
Regional	1.95	1.79–2.13	**<0.001**
Distant	3.15	2.88–3.45	**<0.001**
**Tumor grade**			
Well differentiated (I)	Reference		
Moderately differentiated (II)	1.15	1.01-1.30	**0.031**
Poorly differentiated (III)	1.59	1.41-1.81	**<0.001**
Undifferentiated (IV)	1.47	1.07-2.03	**0.018**

AIAN, American Indian/Alaska Native; aHR, adjusted hazard ratio; API, Asian or Pacific Islander; CI, confidence interval; ICC, intrahepatic cholangiocarcinoma; NH, non-Hispanic. ‡ Adjusted hazard ratio refers to risk of disease-specific mortality risk after adjusting for important covariates. Significant *p*-values (<0.05) are in bold.

## Data Availability

The data supporting the findings of this research are publicly available through the SEER database, accessible through the SEER*Stat software, Version 9.0.41.

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
