# Peer review of "Rural–Urban Disparities in Treatment and Disease-Specific Survival for Patients with Intrahepatic Cholangiocarcinoma: A Retrospective Cohort Analysis of the 2000 to 2021 SEER Database"

_medsci, 2025, doi:10.3390/medsci13030158_

Round 1

Reviewer 1 Report

Comments and Suggestions for Authors

Dear authors,

I read your study with great interest on one side, but on the other side, it’s almost unimaginable to comprehend those differences from my perspective, coming from EU-based healthcare systems. Unfortunately, we see a similar trend in my country, where poorer people find it difficult to get the best possible treatment, even when the treatment is covered by national insurance and there is no need to pay. It’s a great study and a reminder for all of us to establish a system that would allow equality and the same access for all patients. As you presented, intrahepatic cholangiocarcinoma does not recognize income, money, or racial differences.

The only thing that might strengthen your paper, if possible, is presenting additional data on whether the patients (urban vs rural) were treated in tertiary, university, or private centers versus non-tertiary hospitals. If you can provide that data, it would be interesting to see if there are any differences in the type of treatment and survival rates.

Congratulations on you study.

Author Response

Hello,

Thank you for taking the time to review our manuscript. We also share an interest in examining how survival and treatment rates among urban and rural patient populations differ between tertiary, university, or private centers and non-tertiary hospitals. Such a comparison would provide valuable insights into the role of hospital resources in treatment access and help determine whether geographic residence or the type of treating facility is the primary driver of observed disparities. Unfortunately, the SEER database used for our analysis does not provide this level of detail. We hope that, as more comprehensive data become available, future research can address this question and support the development of healthcare policies aimed at strengthening patient navigation programs and expanding investment in regional cancer centers.

Reviewer 2 Report

Comments and Suggestions for Authors

The manuscript entitled " Rural-Urban Disparities in Treatment and Disease-Specific Survival for Patients with Intrahepatic Cholangiocarcinoma: A Retrospective Cohort Analysis of the 2000 to 2021 SEER Database” was evaluated. This study s addresses a significant public health issue by examining rural-urban disparities in treatment patterns and disease-specific survival (DSS) for intrahepatic cholangiocarcinoma (ICC) patients using the SEER database (2000–2021). The manuscript is well structured, the analysis is methodologically sound, and the conclusions highlight critical inequities in cancer care.  The paper is in the scope of the journal and may be published.

Comments:

1.Abstract:

Results: Specify the absolute number of rural/urban patients receiving chemotherapy/surgery (e.g., "rural: 789/1,588 vs. urban: 7,112/13,352").

Conclusion: Replace "crucial" with "warranted" for academic tone.

2.Table 2:

Income: Rename "<$74,999" to "Lower income" and "$75,000+" to "Higher income" for clarity.

3.Discussion:

Stage at Diagnosis: Reiterate that identical stage distribution (Table 1) rules out diagnostic delays as the primary driver—strengthening the argument for treatment-access inequities.

4.References: Ref 10 and 11 are identical (Levit et al. 2020). Correct duplication and diversify supporting literature.

Author Response

Please see attachment below. Thank you.
